# Symbiotic bacteria confer insecticide resistance by metabolizing buprofezin in the brown planthopper, *Nilaparvata lugens* (Stål)

**Bin Zeng**[1,2,3], **Fan Zhang**[1,2], **Ya-Ting Liu**[1,2], **Shun-Fan Wu**[1,2], **Chris Bass**[3]*, **Cong-Fen Gao**[1,2]*

**1** College of Plant Protection, Nanjing Agricultural University, Nanjing, People's Republic of China, **2** State & Local Joint Engineering Research Center of Green Pesticide Invention and Application, Jiangsu, People's Republic of China, **3** College of Life and Environmental Sciences, Biosciences, University of Exeter, Penryn Campus, Penryn, Cornwall, United Kingdom

* c.bass@exeter.ac.uk (CB); gaocongfen@njau.edu.cn (C-FG)

**Data Availability Statement:** The authors confirm that all data underlying the findings are fully

## Abstract

Buprofezin, a chitin synthesis inhibitor, is widely used to control several economically important insect crop pests. However, the overuse of buprofezin has led to the evolution of resistance and exposed off-target organisms present in agri-environments to this compound. As many as six different strains of bacteria isolated from these environments have been shown to degrade buprofezin. However, whether insects can acquire these buprofezin-degrading bacteria from soil and enhance their own resistance to buprofezin remains unknown. Here we show that field strains of the brown planthopper, *Nilaparvata lugens*, have acquired a symbiotic bacteria, occurring naturally in soil and water, that provides them with resistance to buprofezin. We isolated a symbiotic bacterium, *Serratia marcescens* (*Bup_Serratia*), from buprofezin-resistant *N. lugens* and showed it has the capacity to degrade buprofezin. Buprofezin-susceptible *N. lugens* inoculated with *Bup_Serratia* became resistant to buprofezin, while antibiotic-treated *N. lugens* became susceptible to this insecticide, confirming the important role of *Bup_Serratia* in resistance. Sequencing of the *Bup_Serratia* genome identified a suite of candidate genes involved in the degradation of buprofezin, that were upregulated upon exposure to buprofezin. Our findings demonstrate that *S. marcescens*, an opportunistic pathogen of humans, can metabolize the insecticide buprofezin and form a mutualistic relationship with *N. lugens* to enhance host resistance to buprofezin. These results provide new insight into the mechanisms underlying insecticide resistance and the interactions between bacteria, insects and insecticides in the environment. From an applied perspective they also have implications for the control of highly damaging crop pests.

## Author summary

The evolution of insect resistance to insecticides represents a major threat to the sustainable control of many of the world's most damaging crop pests. To effectively combat resistance it is important to understand the mechanisms underpinning resistance and their

available without restriction. All relevant data are within the paper, its Supporting Information files and genome sequencing data generated in this study have been submitted to the NCBI Nucleotide Database under accession number CP097900. (https://www.ncbi.nlm.nih.gov/search/all/?term=CP097900+).

**Funding:** This research was supported by the National Natural Science Foundation of China (No. 31972298 & 32172449 to CG) and the Postgraduate Research & Practice Innovation Program of Jiangsu Province (KYCX21_0626 to BZ). The funders had no role in study design, data collection and analysis, decision to publish, or preparation of the manuscript.

**Competing interests:** The authors declare that no competing interests exist.

contribution to phenotype. Numerous studies have shown that resistance can result from mutations in the insect genome that alter the expression of detoxification enzymes or enhance their activity, or modify the affinity of the insecticide for the target receptor. However, emerging evidence suggests that symbiotic bacteria may also mediate insecticide resistance in host insects. In this study we show that a bacterium commonly found in the environment can be acquired by the economically important pest insect, *Nilaparvata lugens*, and confer resistance to the insecticide buprofezin. Using genomic and transcriptomic analyses we implicate a Rieske nonheme iron oxygenase (RHO) system and VOC family protein encoded in the *Bup_Serratia* genome in the degradation pathway of buprofezin. Our studies uncover a new mechanism of *N. lugens* resistance to buprofezin and illustrate the importance of considering microbe—host insect—environment interactions in the development of integrated pest management strategies.

## Introduction

Crop pests and pathogens cause substantial economic losses ranging from 10%~40% annually for the five major crops (wheat, rice potato, maize and soybean) [1]. Insect pests are a particularly damaging group of crop pests causing damage by feeding and/or via the transmission of plant pathogens such as viruses [2]. As a result, chemical insecticides have been widely used to control insect pests in agricultural production. However, the intensive use of insecticides has led to the evolution of insect resistance to these compounds, severely threatening global crop protection and human food security.

Insect pests have been shown to evolve resistance by a variety of different mechanisms. These include: i) reduced penetration of insecticides, ii) reduced affinity of the target site for the insecticide, and iii) enhanced expression or activity of metabolic enzymes [3,4]. In most cases, insects acquire the ability to resist insecticides from one or more mutations in their genomes. However, recent studies have revealed that symbiotic bacteria can also play a pivotal role in conferring insect host resistance via both direct and indirect routes [5]. For example, some symbiotic bacteria, belonging to the *Burkholderia*, *Citrobacter*, and *Serratia* genera directly confer resistance by breaking down insecticides using detoxification enzymes encoded in their genomes [6–8]. Alternatively, other intracellular bacteria of the *Wolbachia* and *Arsenophonus* genera have been shown to indirectly lead to insecticide resistance by upregulating the expression of P450 detoxification enzyme genes [9–11].

Buprofezin is an insect growth regulator and has excellent efficacy against hemipteran pests such as the damaging rice pest brown planthopper, *Nilaparvata lugens*. Due to the intensive use of buprofezin in the field, since 2013, *N. lugens* has developed a high level of resistance to this compound [12]. Several studies have shown that resistance mechanisms encoded by *N. lugens* own genome can confer resistance to buprofezin, including the overexpression of P450 genes and target site mutation [12–14]. However, in addition to the selective force of insecticides on insect pests, intensive insecticide application can also expose soil bacteria to these compounds and may select for enhanced insecticide biodegradation in the environment [15,16]. Related to this, some insects, such as the bean bug *Riptortus pedestris*, have been shown to acquire pesticide-degrading bacteria from the environment and form a mutualistic relationship with these microbes [8]. Previous studies have found that several different genera of bacteria from the soil are capable of degrading buprofezin [17–19], and the critical genes involved in metabolizing buprofezin have been identified [20]. It is thus feasible that *N. lugens* could acquire such bacteria due to their fitness enhancing effects in the presence of buprofezin.

However, to date the potential of symbiotic bacteria to mediate resistance in *N. lugens* remains unknown.

Here, we used genomic analyses in conjunction with classic culture-dependent experiments to address this knowledge gap. Our analyses addressed the following key questions: 1) Does insecticide exposure modulate the diversity and abundance of gut bacteria in *N. lugens*? 2) Can symbiotic bacteria mediate resistance to the insecticide buprofezin in *N. lugens*? 3) What is the genetic basis for symbiont-mediated degradation of buprofezin (e.g. the key genes encoded in bacterial genomes that confer resistance)?

## Materials and methods

### Insects

The susceptible strain (NIS) of *N. lugens* was collected from Hangzhou, Zhejiang province in 1995 and has been maintained in the lab without exposure to any insecticides since then, its $LC_{50}$ value when tested against buprofezin was 0.93 mg/L. The buprofezin-resistant strain (NIB) originally was collected from Mangshi, Yunnan province in 2020, its $LC_{50}$ value to buprofezin was 41.20 mg/L and then it was successively selected with 60 mg/L buprofezin for five generations. All BPH strains were reared on rice seedlings under standard conditions of $27 \pm 1°C$ and 70–80% relative humidity with a 16-h light/8-h dark photoperiod.

### Selection of *N. lugens* with buprofezin

Third-instar nymphs of *N. lugens* were selected with buprofezin using the rice-seeding dipping method as described previously [21] but with slight modifications. Briefly, about 30 seven-day-old rice seedlings were wrapped together and immersed in 60 mg/L buprofezin solution for 30 seconds and then air-dried at room temperature. Subsequently rice seedlings were placed into a disposable plastic cup. A total of 40 third instar-old nymphs were transferred into each cup and then maintained under the rearing conditions detailed above for 120 h. Twenty-five cups, a total of 1000 nymphs of *N. lugens*, were selected each time. The survival rate was recorded, and survivors were transferred to fresh insecticide-free seedlings to complete the life cycle in preparation for the selection of the next generation.

### Determination of the effect of bacterial symbionts on insecticide susceptibility

Approximately 200 adult *N. lugens* were transferred to 15-day-old rice seedlings that were planted at the bottom of a 1000 mL glass breaker in a layer of absorbent cotton. Adult *N. lugens* were removed from the breaker after five days. To eliminate bacterial symbionts in *N. lugens*, rice seedlings were irrigated with a 100 mL solution of 200 mg/L tetracycline from the first day that adult *N. lugens* were introduced into the beaker, with daily replacement of the tetracycline solution. For the control group, the rice seedlings were irrigated with sterilized water without tetracycline. After about thirteen days, most of the eggs hatched into nymphs which were transferred to fresh rice seedlings without antibiotics to continuously rear until the third instar for buprofezin susceptibility testing.

For insecticide susceptibility testing, seven-day-old rice seedlings were immersed in the insecticide solution for 30 seconds and subsequently air-dried at room temperature. Two robust rice seedlings were transferred to a transparent plastic tube (2.5 cm diameter, 10 cm height) using a sterilized tweezer. A total of 120 third instar nymphs of control and treated groups were chosen randomly and transferred individually into plastic tubes. These were placed in standard conditions for rearing and mortality was recorded each 12 h. Technical

grade buprofezin (98%) and imidacloprid (95%) were provided by Changlong Chemical Industrial Group Co. Ltd. (Jiangsu, China).

## 16S rDNA amplicon sequencing

Fifth-instar *N. lugens* nymphs were used for sample preparation. The insects were surface-washed for 3 minutes with 75% ethanol three times and rinsed five times with sterilized deionized water. Subsequently, DNA was extracted from whole bodies using the HiPure Soil DNA Kits (Magen, Guangzhou, China) according to the manufacturer's protocols. For 16S rDNA Illumina sequencing, each treatment had three independent biological replicates and each replication contained twenty *N. lugens* nymphs.

A 465 bp amplicon of the V3-V4 hypervariable region of the bacterial 16S rDNA gene was amplified by PCR(95˚C for 2 min, followed by 35 cycles of 95˚C for 30 s, 60˚C for 45 s, and 72˚C for 90s, with a final extension 72˚C for 10 min)using specific primers to which a barcode was added. The primer sequences were as follows: 341F (5'-CCTACGGGNGGCWGCAG-3') and 806R (5'-GGACTACHVGGGTATCTAAT-3'). PCR reactions were carried out in a 50 μL reaction volume with TransGen High-Fidelity PCR SuperMix (TransGen Biotech, Beijing, China), 0.2 μM forward and reverse primers, and 5 ng template DNA. The resulting PCR products were run on 2% agarose gels and purified using the AxyPrep DNA Gel Extraction Kit (Axygen Biosciences, Union City, CA, USA) according to the manufacturer's instructions and quantified using the ABI StepOnePlus Real-Time PCR System (Life Technologies, Foster City, USA). Purified amplicons were pooled in equimolar and paired-end sequenced (PE250) on an Illumina platform according to the standard protocols.

## Bioinformatic analyses

Raw reads were filtered by removing reads containing more than 10% of unknown nucleotides and low-quality reads containing less than 50% of bases with quality (Q-value)>20 using FASTP [22] (version 0.18.0) to generate high-quality clean reads. Paired-end clean reads were merged as raw tags using FLASH [23] (version 1.2.11) with a minimum overlap of 10 bp and mismatch error rates of 2%. Noisy sequences in the raw tags were filtered under specific filtering conditions [24] to obtain high-quality clean tags. The clean tags were clustered into operational taxonomic units (OTUs) of ≥ 97% similarity using the UPARSE [25] (version 9.2.64) pipeline. All chimeric tags were removed using the UCHIME algorithm [26] to generate a final set of tags for further analysis. The tag sequence with the highest abundance was selected as representative sequence within each cluster. The representative OTU sequences were classified into organisms by a naive Bayesian model based on the RDP classifier [27] (version 2.2) and the SILVA database [28] (version 138.1) with a confidence threshold value of 0.8. Alpha diversity analysis was performed in QIIME [29] (version 1.9.1). The difference comparison between NIS and NIB was calculated by Welch's t-test in the R project Vegan package [30] (version 2.5.3). Jaccard and Bray-Curtis distance matrix were calculated using the R project Vegan package [30] (version 2.5.3). PCA (principal component analysis) was performed in the R project Vegan package [30] (version 2.5.3). The stacked bar plot of community composition was visualized in the R project ggplot2 package [31] (version 2.2.1).

## *Bup_Serratia* isolation and identification

Growth media were prepared as follows. Luria-Bertani (LB) medium contained the following ingredients (g/L): tryptone 10.0, yeast extract 5.0, and NaCl 5.0 (PH: 7.0–7.5). Mineral salts medium (MM) contained the following salts (g/L): K2HPO4 1.5, KH2PO4 0.5, NH4NO3 1.0,

MgSO4·7H2O 0.2, NaCl 1.0 (PH: 7.0–7.5). Buprofezin (50 mg/L) was added to MM to make up the BMM medium. Solid media plates were prepared by adding 2% agar.

Ten fifth-instar nymphs from the NIB *N. lugens* strain were collected and soaked in 70% ethanol for 3 min and then rinsed five times with sterilized deionized water to remove surface bacteria. Subsequently, they were homogenized in a single sterilized tube with 1 mL of steril-ized deionized water. Homogenized samples were centrifuged at 5000 rcf/min for 5 min. The supernatant was added to a 250 mL flask containing 100 mL of BMM and incubated at 200 rpm/min, 30°C for 7 days, where buprofezin was used as the sole carbon source. Approxi-mately 5% of the enrichment culture was inoculated to 100 mL identical fresh BMM every 7 days for three times to obtain pure cultures. Finally, the pure enrichment culture was diluted to $10^{-7}$ concentrations by ten-fold gradient dilutions. 0.2 mL of the dilution was inoculated onto BMM agar plates and incubated at 30°C for several days, with daily monitoring of growth. Ten colonies with the same morphology were randomly picked for subculturing in liq-uid BMM. 16S rDNA of each cultivated bacteria strain was amplified with the universal prim-ers: 27F (5'-AGAGTTTGATCCTGGCTCAG-3′) and 1492R (5'-TACGGCTACCTTGTTAC GACTT-3′) and sequenced. BLAST analysis of the sequences obtained was performed against the National Center for Biotechnology Information (NCBI) database. Phylogenetic trees were generated based on the 16S rDNA sequence using PhyML with WGA as the substitution model and 1000 bootstraps in Geneious (version 10.2.6, Biomatters, New Zealand).

## Identification of *Bup_Serratia* degrading buprofezin

*Bup_Serratia* was precultured in LB medium at 30°C, 220 rpm/min until the optical density at 600 nm ($OD_{600}$) reached 0.8–1.0. Cells were harvested by centrifuging at 4°C and 5000 rpm/min for 5 min and washed twice with sterilized deionized water. The cells were resuspended and the $OD_{600}$ value was adjusted to 2.0 with the BMM containing buprofezin of 50 mg/L. Next, the cells were inoculated at 5% (v/v) into a 100 mL flask containing 20 mL of BMM and incubated at 220 rpm/min and 30°C for 24 h. To extract residual buprofezin, isopyknic dichloromethane was added to a 3 mL culture solution and oscillated thoroughly for 1 min. The extraction mixture was centrifugated at 12000 rpm/min for 1 min to separate the organic phase from the water phase. The sub-organic phase was collected and evaporated at 45°C using a termovap sample concentrator. The product was dissolved in 0.5 mL methanol and fil-tered through a 0.22-um membrane filter. The amount of residual buprofezin was immediately determined by reverse-phase HPLC with a $C_{18}$ column (Ameritech Technology). The mobile phase was methanol: water (80: 20 v/v) with a flow rate of 0.8 ml min$^{-1}$ at 40°C. The detection wavelength was 230 nm and the loading volume was 20 μL. BMM without *Bup_Serratia* was used as the control. Three replicates were prepared for each treatment. The buprofezin stan-dard sample served as a control to identify the retention time of buprofezin.

## Infection of *N. lugens* with *Bup_Serratia*

About 200 adult *N. lugens* from the NIS strain were transferred to 15-day-old rice seedlings that were planted in a 1000 mL glass breaker on a layer of absorbent cotton. After five days adult *N. lugens* will be removed from the breaker. 100 mL of *Bup_Serratia*, which was cultured in LB medium and grown to an $OD_{600}$ of 0.8–1.0, was poured into the beaker on the first day that *N. lugens* were transferred. *Bup_Serratia* was replaced with fresh bacteria every three days. After approximately thirteen days, the majority of the eggs had hatched into nymphs. These nymphs were then transferred to fresh rice seedlings that were free from *Bup_Serratia* for con-tinuous rearing for five days until they reached the third instar, after which they were used for

buprofezin susceptibility testing as described above. Both LB medium without *Bup_Serratia* and containing *Escherichia coli* were used as a control.

## *Bup_Serratia* genome sequencing, annotation and identification of buprofezin degrading genes

*Bup_Serratia* was cultured in LB media at 30°C until the $OD_{600}$ reached about 0.8–1.0. Genomic DNA was extracted from liquid cultures using HiPure bacterial DNA Kits (Magen, Guangzhou, China) according to the manufacturer's instructions. The quality of genomic DNA was assessed using a Qubit (Thermo Fisher Scientific, Waltham, MA) and by gel electrophoresis. A combination of PacBio sequencing and Illumina sequencing was performed by Genedenovo Biotechnology Co., Ltd (Guangzhou, China). Continuous long reads from PacBio sequencing were *de novo* assembled using Falcon (version 0.3.0) [22], and clean reads from Illumina sequencing were used to polish the genome sequence and improve the quality of the final assembly. The final genome assembly has been deposited in GenBank under the accession number CP097900.

ORFs (Open reading frames) were predicted using Prokka version 1.11 [32] with default parameters. Noncoding RNAs such as rRNAs, tRNAs and sRNAs were predicted and identified by rRNAmmer [33] (version 1.2), tRNAsacn [34] (version 1.3.1) and cmscan [35] (version 1.1.2). Functional annotation of assembled genes was performed by comparison against several databases including the NCBI non-redundant protein sequence (Nr) database, UniProt/Swiss-Prot, Kyoto Encyclopedia of Genes and Genomes (KEGG), Gene Ontology (GO), Cluster of Orthologous Groups of proteins (COG). Protein family annotation was applied with Pfam_Scan (version 1.6) [36] based on the Pfam database (version 32.0). Syntenic analyses were performed using Mauve (version 2.4.0) [37]. A circular map of the genome was generated using Circos version 0.64 [38]. To identify which genes were involved in the degrading of buprofezin in *Bup_Serratia*, the nucleotide and protein sequences of *Rhodococcus qingshengii* YL-1 *BfzA1A3A4*, *BfzB* and *BfzC* (KY785168.1) that have been shown to be involved in degrading buprofezin [20] were used as the queries sequences to perform local BLAST searches against the nucleotide and protein sequence databases of *Bup_Serratia*.

## Expression analysis of candidate genes putatively involved in buprofezin degradation by real-time qPCR

*Bup_Serratia* was cultivated in BMM media containing 0, 5 mg/L, 12.5 mg/L, 25 mg/L and 50 mg/L of buprofezin for 48 h, respectively. Total RNA was extracted using the Bacteria RNA Extraction Kit (Vazyme, Nanjing, China) following the manual. First-strand cDNA was reverse transcribed using HiScript III RT SuperMix for qPCR (+gDNA wiper) (Vazyme, Nanjing, China) following the manufacturer's protocol. The *rplU* gene was used as a housekeeping gene for normalisation [39]. Real-time quantitative PCR amplification was performed on the Applied Biosystems 7500 Real-Time PCR System (Applied Biosystems by Life Technologies, Carlsbad, CA) using AceQ Universal SYBR qPCR Master Mix (Vazyme, Nanjing, China) according to the manufacturer's protocol. The amplification efficiency of all primers (S1 Table) was assessed using a serial dilution of 100 ng to 0.01 ng of cDNA.

## Detection of *Bup_Serratia* in different field populations of *N. lugens*

To detect the prevalence of infection with *Bup_Serratia* in different field populations of *N. lugens*, 10 populations (S2 Table) collected in 2022 were subjected to real-time quantitative PCR against the reference gene of *rplU* to obtain the cycle threshold (Ct) value. A 191 bp

region of the *rplU* gene in *Bup_Serratia* was amplified with the specific primers (S1 Table) on the Applied Biosystems 7500 Real-Time PCR System (Applied Biosystems by Life Technologies, Carlsbad, CA) using AceQ Universal SYBR qPCR Master Mix (Vazyme, Nanjing, China) according to the protocol.

### Statistical analysis

*N. lugens* survival rate was analyzed using the log-rank (Mantel-Cox) test. The richness index of alpha and beta diversity between NIS and NIB strains was compared by student's *t*-test. The relative expression levels of genes of interest were analyzed by one-ANOVA and Tukey's multiple comparisons test. Differences were considered significant if the *p* values were < 0.05. All the statistics were conducted on GraphPad Prism 7.0.

## Results

### Bacterial symbionts are associated with buprofezin resistance in *N. lugens*

A field population of *N. lugens* collected in 2020 from Mangshi, Yunnan province in China was selected with 60 mg/L of buprofezin for five generations, in order to enrich for the presence of any buprofezin-degrading bacteria. A significant increase in survival was observed over the selection period, from just 16% of the G0 generation to 72% of the G5 (NIB) generation (Fig 1A). Interestingly, survival to buprofezin (60 mg/L) significantly decreased in the NIB strain after treatment with the antibiotic tetracycline (Fig 1B), however, exposure to this antibiotic had no effect on the sensitivity of the NIB strain to imidacloprid (an insecticide with a different mode of action) (Fig 1C). Furthermore, when the insecticide susceptible strain NIS strain was treated with tetracycline no change in buprofezin susceptibility was observed (Fig 1D). These results provided initial evidence that the enhanced tolerance of the NIB strain to buprofezin may involve symbiotic bacteria.

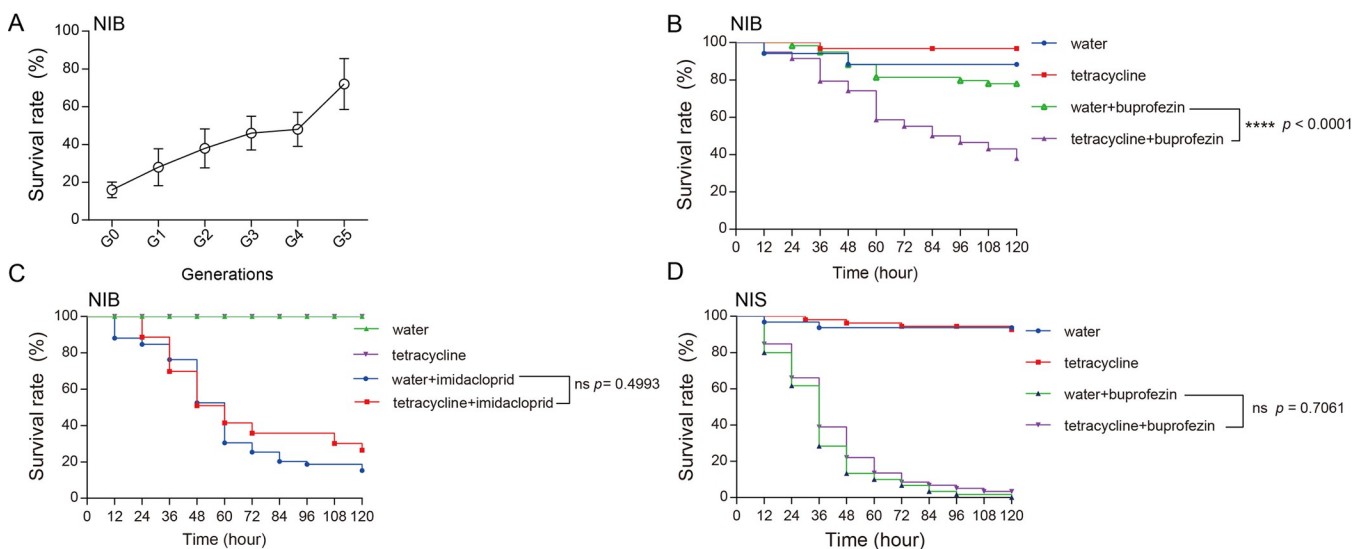

**Fig 1. The buprofezin resistance may be associated with symbiont bacteria in the *N.lugens*.** (A) The survival increased across generations after being selected with 60 mg/L buprofezin in each generation. (B) The effect of disrupting symbiont bacteria on survival following buprofezin exposure in the resistant strain. (C) The effect of disrupting symbiont bacteria on survival following imidacloprid exposure in the resistant strain. (D) The effect of disrupting symbiont bacteria on survival following buprofezin exposure in the susceptible strain. Difference comparisons were performed in GraphPad 7.0 with the log-rank test. ****, indicate *p* < 0.0001; ns, no significant.

## Analysis of the microbiome of the NIS and the NIB *N. lugens* strains

To assess bacterial composition and diversity in the NIS and NIR strains, we sequenced the variable region of the 16S rDNA gene using high-throughput amplicon sequencing. As shown in the supplementary file: S3 Table, more than 130000 raw reads were generated from each sample of the NIS and NIB strains, and more than 110000 effective tags were obtained from each sample after raw reads were filtered and chimeras removed. The ratio of effective tags was more than 88% in all samples and the value of Good's coverage close to 1, indicating that sequencing quality and depth were sufficient for effective downstream analysis.

Alpha diversity analysis revealed that Sob, Chao1 and ACE indices showed no significant difference between the NIS and NIB strains (S1 Fig). In contrast, Shannon and Simpson indices, which reflect species evenness and richness respectively, differed markedly between the NIS and NIB strains (Fig 2A). Similarly, significant differences were observed in beta diversity and Principal coordinate analysis (PCoA) between the NIS and NIB strains (Fig 2B and 2C). Together, these results reveal significant differences in the bacteria community composition

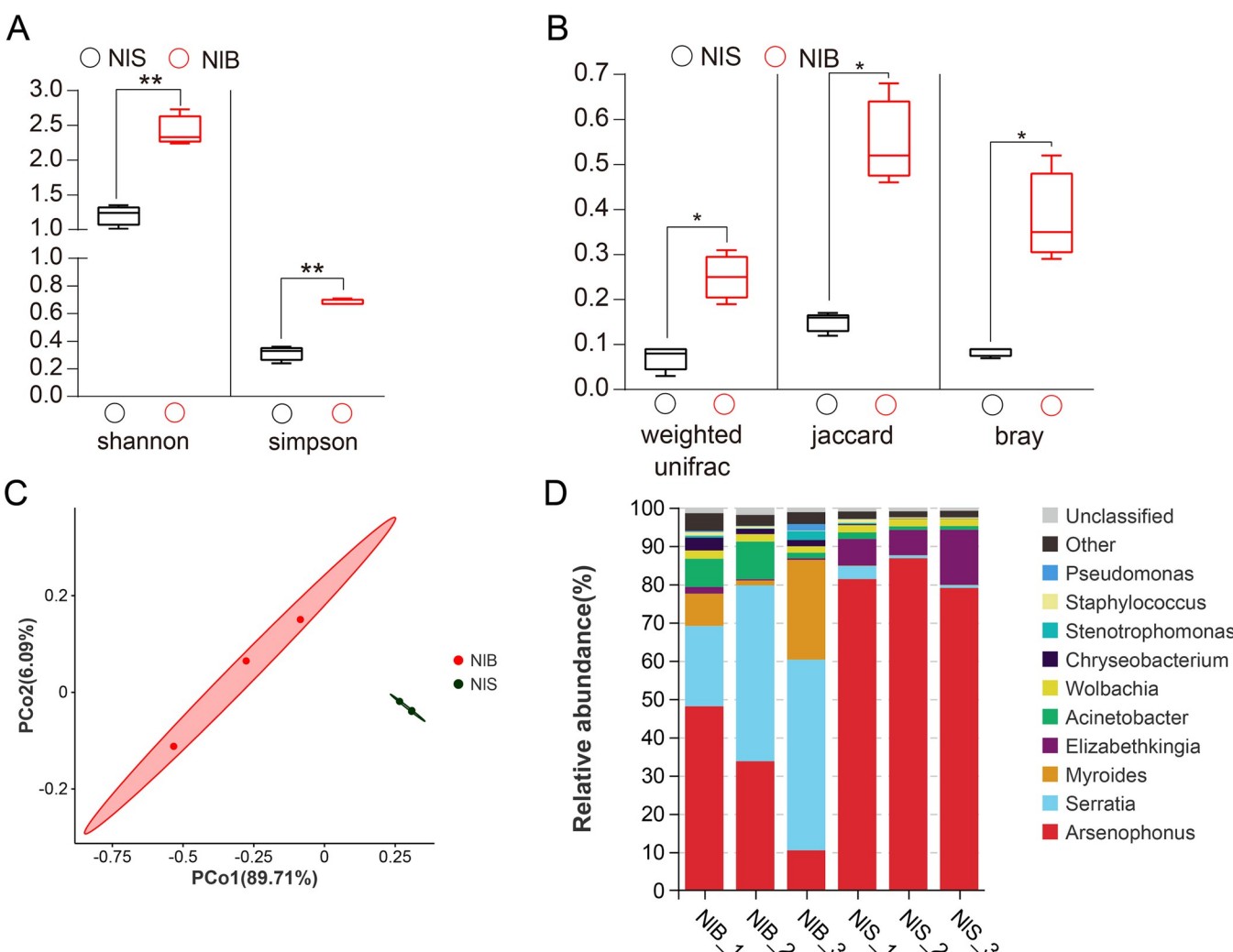

**Fig 2. Different symbiotic bacterial communities between resistant and susceptible strains of *N. lugens*.** (A) Alpha diversity analysis between resistant and susceptible strains of *N. lugens*. (B) Beta diversity analysis between resistant and susceptible strains of *N. lugens*. (C) Principal coordinate analysis (PCoA) plots of Bray–Curtis distances for samples' bacterial communities. Each point corresponds to a different sample; different colors correspond to different treatments. (D) A stack map at the genus level for six samples. Significant differences are indicated by asterisks: $*p < 0.05$, $**p < 0.01$ (Kruskal-Wallis test).

and structure between buprofezin resistant and susceptible strains of *N. lugens*. Most notably, we observed a significantly higher abundance of bacteria of the *Serratia* genus in the NIB strain compared to the NIS strain (Fig 2D).

## Isolation and identification of *Bup_Serratia*

Carbon and nitrogen are indispensable for bacterial growth. We therefore exploited MM containing buprofezin as a sole carbon source to isolate bacteria from the NIB strain which have the capacity to utilize this insecticide. A bacterial strain was identified that showed excellent growth on liquid and solid media containing 50 mg/L buprofezin but was unable to grow on MM without buprofezin (Fig 3A and 3B). PCR and sequencing of a 16S rDNA gene fragment of 1405 bp from this strain resulted in a sequence with 99.5% identity to *Serratia marcescens* ATCC 13880 upon interrogation against the EzBioCloud database (https://www.ezbiocloud. net/). We named the strain *Bup_Serratia* (Fig 3C) and deposited it in GenBank under accession number OQ345816.

## *Bup_Serratia* confers buprofezin resistance to its host by metabolizing buprofezin

To determine whether the *Bup_Serratia* could confer buprofezin resistance to *N. lugens*, the NIS strain was irrigated with LB media containing *Bup_Serratia* at an $OD_{600}$ of 2.0, or LB media without *Bup_Serratia* as a control. A significant increase in resistance to buprofezin was observed for the NIS strain inoculated with *Bup_Serratia* compared to the control (Fig 3D), however, there was no change in sensitivity to buprofezin after the NIS strain was inoculated with *E. coli* compared to the control (S2 Fig). To further confirm the role of *Bup_Serratia* in degrading buprofezin, *Bup_Serratia* was added to BMM that contained 50 mg/L buprofezin, and High-Performance Liquid Chromatography analysis of the metabolic fate of buprofezin

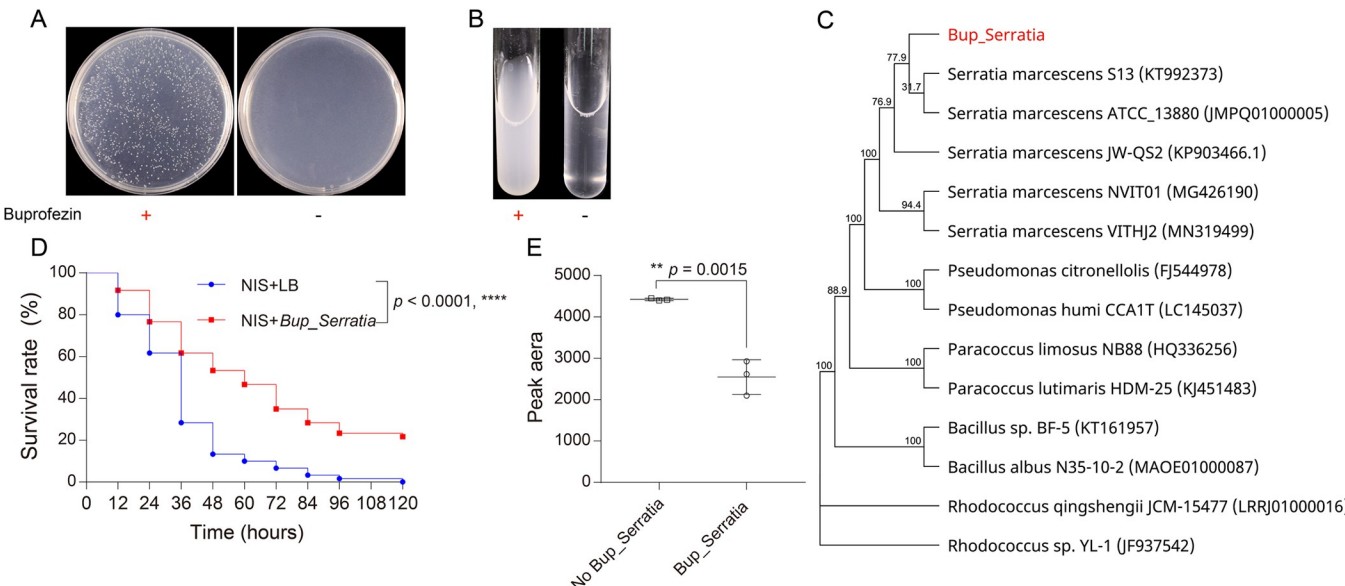

**Fig 3. Bacteria *Bup_Serratia* isolated from *N. lugens* can metabolize buprofezin and increase host resistance.** The growth comparison of *Bup_Serratia* between MM containing 50 mg/L buprofezin and only MM, (A) solid and (B) liquid medium. (C) Phylogenetic relationship of the symbiotic *Bup_Serratia* strain with a maximum likelihood based on 16S rRNA sequences. (D) The sensitivity changes of *N. lugens* to buprofezin after *Bup_Serratia* was incubated to NIS strain. Difference comparisons were performed in GraphPad 7.0 with the log-rank test. ****, indicate P < 0.0001. (E) The quantity determination of buprofezin by HPLC after buprofezin was incubated with *Bup_Serratia* in MM medium for 24 h. Significant differences are indicated by asterisks: **$p < 0.01$ (t-test).

was conducted. This revealed that the parent buprofezin significantly decreased in quantity in BMM containing *Bup_Serratia* compared with media lacking *Bup_Serratin* (Fig 3E). Taken together, these results provide unequivocal evidence that *Bup_Serratia* has the capacity to directly metabolize buprofezin and confer resistance to this compound in *N. lugens*.

## Whole genome sequence analysis of *Bup_Serratia* and identification of buprofezin-degrading genes

Several bacteria genera from soil have been shown to have the capacity to metabolize buprofezin, such as *Rhodococcus* (sp) YL-1 [19], and the key genes involved have been characterized. To understand the genetic basis of buprofezin metabolism by *Bup_Serratia*, we performed whole genome sequencing of this bacteria using a combination of PacBio sequencing and Illumina sequencing. The complete genome assembly of *Bup_Serratia* comprises a single scaffold of 5.11Mb with an average GC content of 59.47%. A total of 4,628 coding sequences (CDS), 88 tRNA, 22 rRNA and 37 sRNA were predicted in the assembly (Fig 4A and S4 Table). Analysis of genomic synteny among *Serratia marcescens* ATCC 13880, *Bup_Serratia*, and a 200 kb fragment of *Rhodococcus* (sp) YL-1 that contains key genes involved in buprofezin metabolism [20], revealed genomic regions that are highly conserved (Fig 4B).

The buprofezin degradation pathway of *Rhodococcus* (sp). YL-1 has been shown to involve the genes *BfzA1A2*, *BfzA3*, *BfzA4*, *BfzB* and *BfzC* encoding metabolic enzymes [20] (Fig 4C). To investigate whether *Bup_Serratia* possesses similar genes, the protein sequences of BfzA1, BfzA2, BfzA3, BfzA4, BfzB and BfzC were used as queries to perform local blastp and tblastn against the *Bup_Serratia* genome and proteome databases. Two sequences, NAE95_20165 and NAE95_20050, were identified in searches with BfzA1, which had 38.7% and 39.2% amino acid sequence identity to this sequence. Homologous sequences were also identified for BfzA3, BfzA4, BfzB and BfzC, comprising NAE95_20175 (75.8% sequence identity), NAE95_20145 (61.9%), NAE95_03695 (61.8%) and NAE95_20150 (55.8%), respectively (Table 1). NAE95_20165 and NAE95_20050 were annotated as aromatic ring-hydroxylating dioxygenase alpha subunits. NAE95_20175, NAE95_20145, NAE95_03695 and NAE95_20150 were annotated as non-heme iron oxygenase ferredoxin subunit, FAD-dependent oxidoreductase, aldo/ keto reductase and VOC family protein respectively. No homologous gene was identified for BfzA2. With the exception of NAE95_20050, phylogenetic analysis grouped these genes into clades with those genes that were involved in the biodegradation of buprofezin in *Rhodococcus* (sp). YL-1 (Fig 4E). When the location of these genes was mapped on the *Bup_Serratia* genome NAE95_20175, NAE95_20165, NAE95_20150 and NAE95_20145 were found to be in close proximity in the genome (Fig 4D) suggesting they may share a generalized function.

*S. marcescens* is widely distributed in environments and comprises diverse strains. To evaluate the extent of conservation of the candidate buprofezin degrading genes identified in *Bup_Serratia* in other strains, we interrogated the genome sequences of three other strains of *S. marcescens*, *S. marcescens* ATC13880 (accession number: CP041234.1), *S. marcescens* BP2 (accession number: CP050013.1) and *S. marcescens* CM2012-028 (accession number: CP091122.1), collected from different ecological niches. This revealed that *NAE95_20050* is not present in the *S. marcescens* CM2012-028 genome. Furthermore, while the sequence of *NAE95_20175* is completely conserved in the examined strains, the encoded amino acid sequences of *NAE95_20165*, *NAE95_20050*, *NAE95_20145*, *NAE95_03695* and *NAE95_20150* vary at 7, 315, 12, 18 and 9 amino acid residues respectively. Thus, the capacity to degrade buprofezin may not be common to all *S. marcescens* strains (S3–S8 Fig).

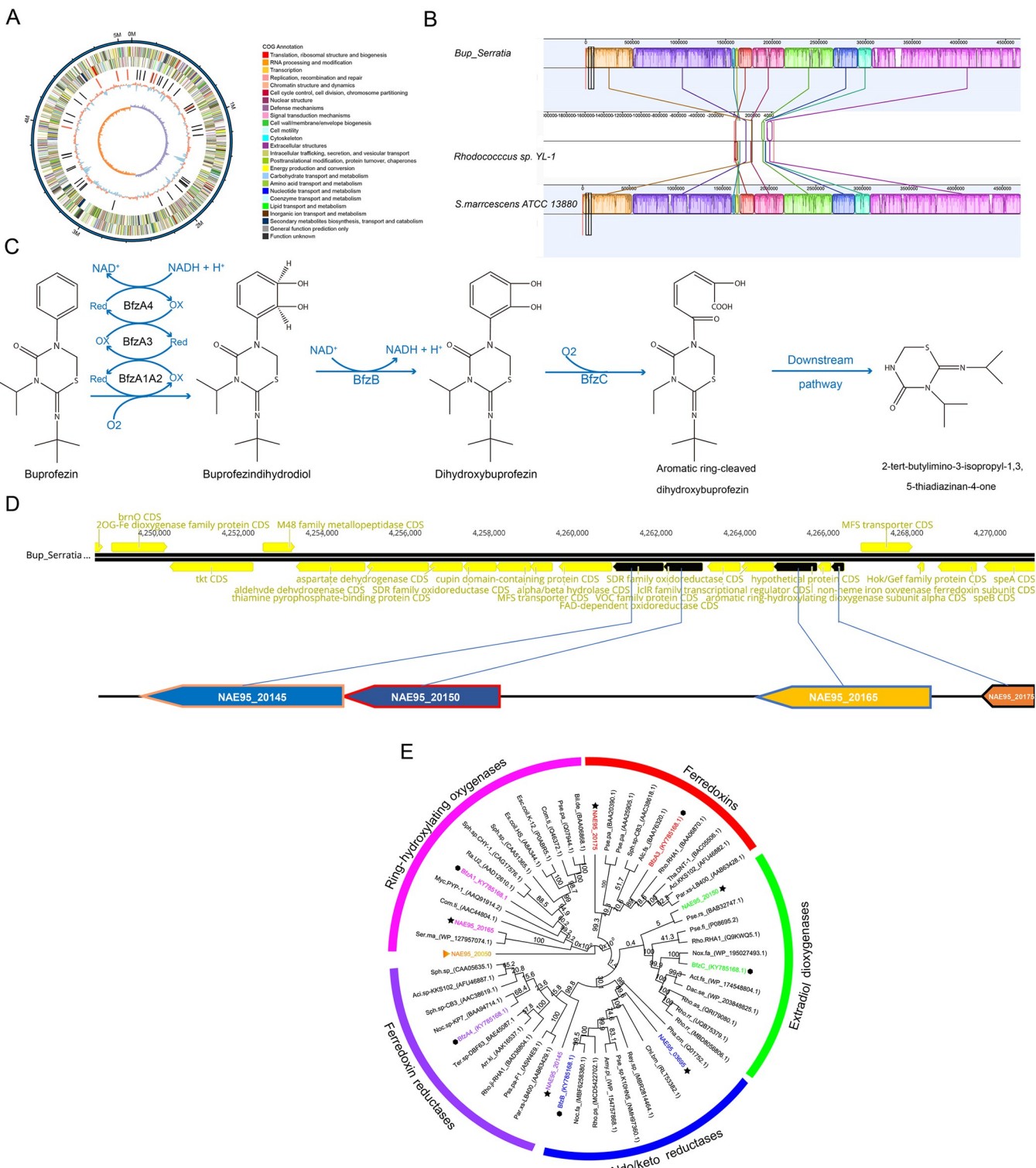

**Fig 4. *Bup_Serratia* genomic characterization and comparative analysis.** (A) Circus map of the genome of the *Bup_Serratia*. The outermost circle is the genome sequence position coordinates. The distribution of the circle from the outside to the inside is the sense strand CDS, antisense strand CDS, ncRNA (black denotes tRNA, red denotes rRNA), GC ratio (red and blue indicate GC ratio is higher and lower than average, respectively), and GC skew, which is used to assess the relative ratio between G and C (GC skew = (G-C/(G+C), where purple is higher than zero and orange is lower than zero). (B) Syntenic relationship between the genome of *Bup_Serratia*, *Rhodococcus* sp. YL-1 and *Serratia marcescens* CM2012_028. (C) The buprofezin metabolizing pathway and identified genes involved in the catabolic pathway in *R. qingshengii* YL-1, referred to by Chen et al [20]. (D) The location of NAE95_20145, NAE95_20150, NAE95_20165

and NAE95_20175 on the genome. (E) Phylogenetic analysis of genes involved in the buprofezin metabolizing pathway. Hexagon indicates these genes have been shown to degrade buprofezin in *R. qingshengii* YL-1. The pentagram indicates these genes are from the *Bup_Serratia* genome. Proteins sequences were aligned using the MUSCLE algorithm in Geneious (version 10.2.6, Biomatters, New Zealand). MEGA 7.0 was used to determine the best-fit model of amino acids substitution. Alignments were used to generate a maximum likelihood tree using PhyML with WGA as the substitution model and 1000 bootstraps. All of the genes information used for phylogenetic construction was listed in S5 Table.

### Induction of gene expression following exposure to buprofezin

To gain additional insight into the candidate buprofezin degrading genes identified in the *Bup_Serratia* genome detailed above we examined their expression response to different concentrations of buprofezin (Fig 5). The results showed the expression levels of *NAE95_20165*, *NAE95_20175*, *NAE95_20145*, and *NAE95_20150* were significantly increased when exposed to buprofezin at 12.5 mg/L to 50 mg/L. The expression of *NAE95_20050* was also significantly upregulated at low concentrations of this insecticide. In contrast, the expression of *NAE95_03695* was not induced by exposure to the four concentrations of buprofezin tested.

### Detection of *Bup_Srratia* in field populations of *N. lugens*

To investigate the prevalence of *Bup_Serratia* symbionts in the field population of *N. lugens*, a quantitative PCR diagnostic based on the *Bup_Serratia rplU* gene was developed and used to screen 10 populations of *N. lugens* collected from diverse regions of China. *Bup_Serratia* was detected in all 10 populations of *N. lugens* tested (Fig 6), demonstrating that *Bup_Serratia* is found at high prevalence in *N. lugens* in the field. However, some variation was observed in the titer of *Bup_Serratia* in different *N. lugens* populations, with two populations from Hunan province, Ningxiang (NX) and Shaodong (SD) exhibiting a higher density of *Bup_Serratia* than the other eight populations.

### Discussion

Our data reveal a symbiotic relationship between a bacteria, that occurs naturally in soil and water, and a damaging insect pest that results in resistance to a key insecticide used for pest control. These findings provide insight into the molecular mechanisms of insecticide resistance and illustrate how interactions between the environment, bacteria and insects can influence the fitness of economically important crop pests, and our ability to control them.

An emerging body of work has shown that the sensitivity of insects to insecticides can be strongly influenced by symbiotic bacteria. These bacteria can enhance insecticide resistance by directly breaking down insecticides or indirectly by regulating the expression of genes encoding detoxification enzymes, and/or by modulating the immune system of the host [7–9,11,40]. In the case of *N. lugens*, previous research has suggested that symbiotic bacteria of the *Wolbachia* and *Arsenophonus* genera can influence the resistance of the host to imidacloprid by

**Table 1. Comparison of buprofezin-metabolizing genes relevant to the current study.**

| BfzA1 | BfzA3 | BfzA4 | BfzB | BfzC | | Annotation information in the genome of *S. marcescens* |
|---|---|---|---|---|---|---|
| 39.2% | | | | | NAE95_20165 | aromatic ring-hydroxylating dioxygenase subunit alpha CDS |
| 38.7% | | | | | NAE95_20050 | ring-hydroxylating oxygenase subunit alpha CDS |
| | 75.8% | | | | NAE95_20175 | non-heme iron oxygenase ferredoxin subunit CDS |
| | | 61.9% | | | NAE95_20145 | FAD-dependent oxidoreductase CDS |
| | | | 61.8% | | NAE95_03695 | aldo/keto reductase CDS |
| | | | | 55.8% | NAE95_20150 | VOC family protein CDS |

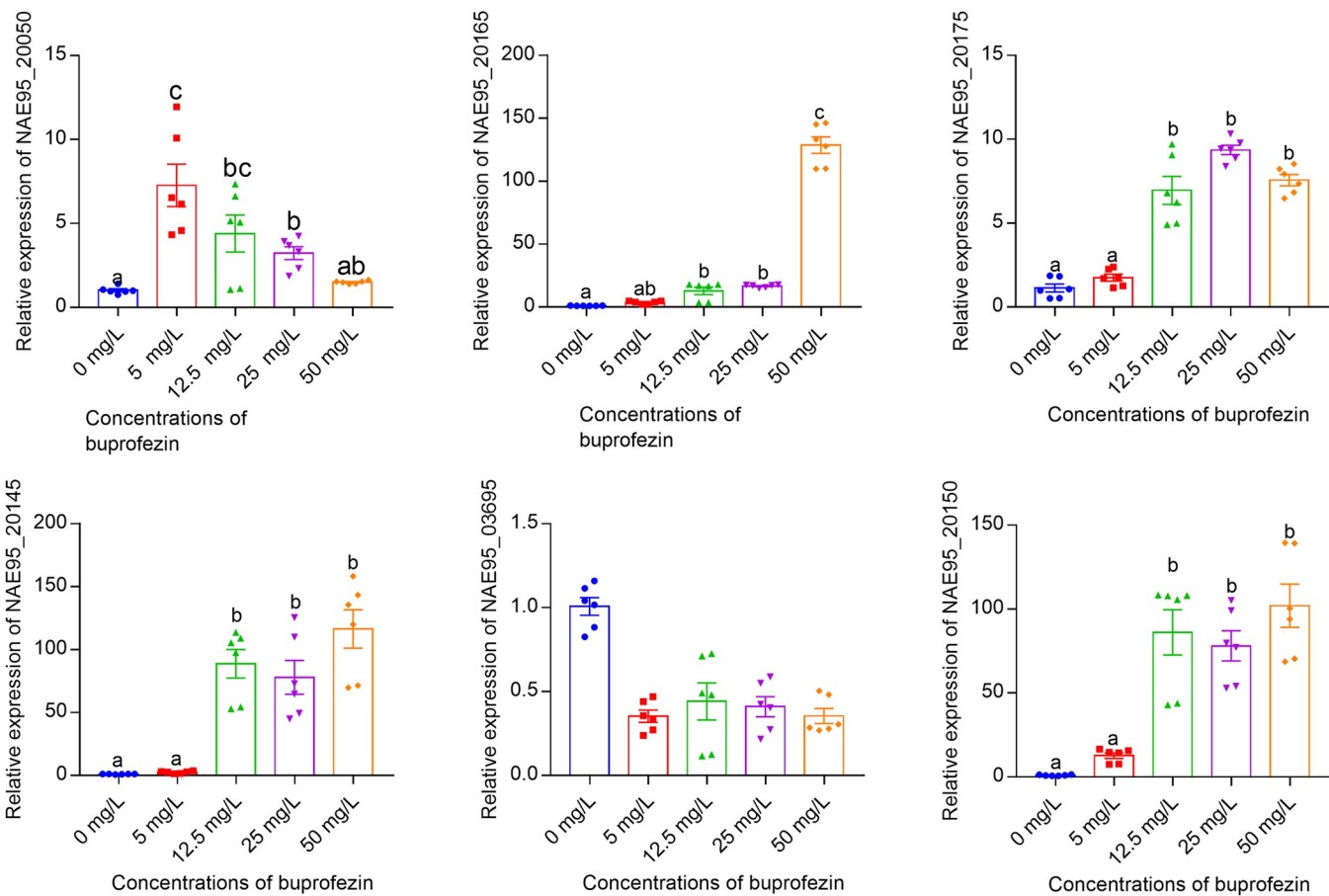

**Fig 5. Relative expression of the six identified homologous genes after exposure to buprofezin.** Expression of these genes was measured against the reference gene *rplU*. Bars (mean ± SE) labeled with the same letter within each treatment are not significantly different ($p > 0.05$, one-ANOVA and Tukey's multiple comparisons test).

regulating the expression of the *N. lugens* P450 genes *CYP6ER1* and *CYP6AY1* [9,11]. However, in this study, we isolated a strain of symbiotic bacteria (*Bup_Serratia*) from a buprofezin-resistant strain of *N. lugens* and provided several lines of evidence that it has the capacity to metabolize buprofezin and thus directly enhance the resistance of its insect host to this insecticide. Specifically, we show: i) The microbiome of the buprofezin-resistant NIB strain of *N. lugens* contains a greater abundance of bacteria of the *Serratia* genus compared to the buprofezin-susceptible NIS strain, ii) the sensitivity of the NIB strain to buprofezin significantly increased after treatment with the antibiotic tetracycline, but this was not the case for the NIS strain, iii) *Bup_Serratia* isolated from the NIB strain can survive on media containing buprofezin as the sole carbon source, iv) the amount of parent buprofezin significantly decreased in media containing this insecticide and *Bup_Serratia* when compared with media containing this insecticide but lacking *Bup_Serratia*, as determined by HPLC analysis, and, v) inoculation of the NIS strain of *N. lugens* with *Bup_Serratia* isolated from the NIB strain enhanced its tolerance to buprofezin. Together, these results provide unequivocal evidence that *Bup_Serratia* has the capacity to directly metabolize buprofezin and confer resistance to this compound in *N. lugens*.

Most symbionts that mediate the detoxification of pesticides are facultative symbionts [7,8,41] that are acquired from the environment. Although previous studies have reported that

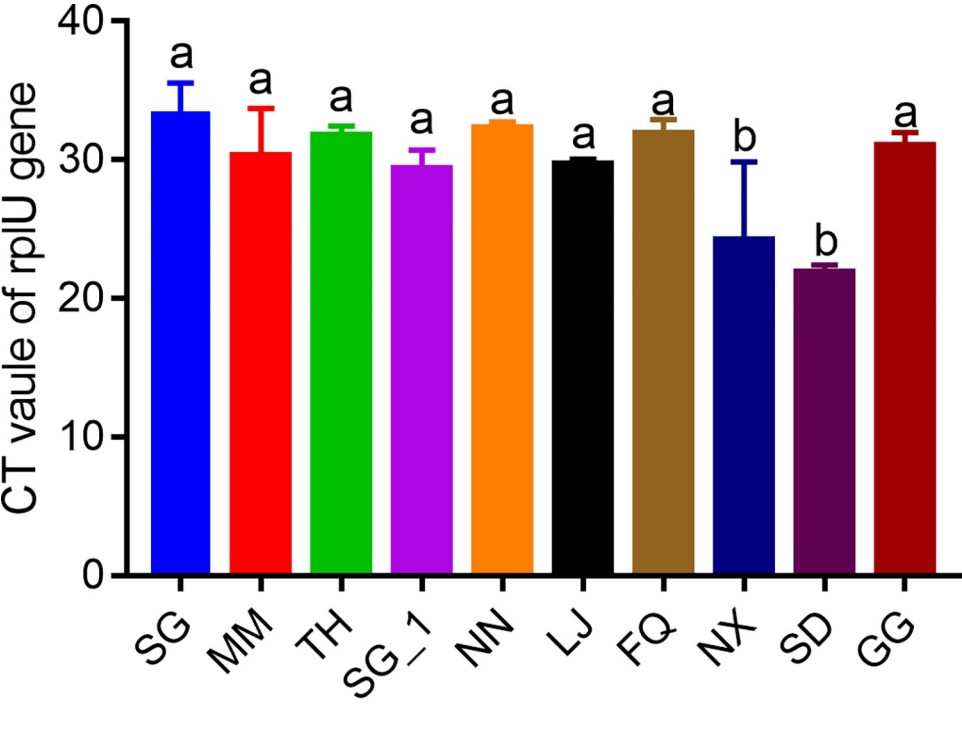

**Fig 6. Detection of *Bup_Serratia* in *N. lugens*** from diverse populations. Bars (mean ± SE) labeled with the same letter within each treatment are not significantly different ($p > 0.05$, Tukey's test).

diverse microorganisms from the environment, including bacteria from the *Paracoccus*, *Pseudomonas*, *Rhodococcus*, and *Bacillus* genera have the capacity to degrade buprofezin as their sole source of carbon and energy for growth [17,19,42,43], none of these bacteria were linked to buprofezin-resistance in our study. Rather, phylogenetic analysis based on 16S rDNA sequences revealed that *Bup_Serratia* is a strain of *S. marcescens*. This strain belong to the family *Enterobacteriaceae*, which is widely distributed in various environments, and is most well known as an opportunistic pathogen of humans, where it usually causes nosocomial infections and outbreaks in severely immunocompromised or critically ill patients [44]. Remarkably, given the findings of our study, *S. marcescens* has also been reported to infect insects and have insecticidal properties against some strains. For example, *S. marcescens* has been shown to produce a Serralysin-like protein with insecticidal activity towards larvae of the scarab beetle *Phyllophaga blanchardi* [45]. Similarly, *S. marcescens* can produce a prodigiosin pigment that is highly virulent to mosquito *Aedes aegypti* larvae [46]. However, *S. marcescens* is an ecologically and genetically diverse bacterial strain [47] and thus strains of *S. marcescens* from different ecological niches may exhibit genotypic differences that modify their pathogenicity to insects [48]. Indeed, as revealed by our analysis of candidate buprofezin degrading genes, although *Bup_Serratia* and the other three *S. marcescens* strains from different ecological niches share many of the same genes, their encoded amino acid sequences have notable variations, which may be expressed in functional differences. Furthermore, previous studies have shown that *S. marcescens can* form mutualistic relationships with insects. Of most relevance to our study, previous work on the wasp, *Nasonia vitripennis* and bean bug *R. pedestris* has shown that *S. marcescens* can utilize the herbicide atrazine and the insecticide dimethoate respectively as the

sole carbon resources for growth and enhance host resistance to these compounds [6,41]. In addition, research on buprofezin resistance in the small brown planthopper infected with *S. marcescens* provided initial evidence that this bacterial strains may enhance resistance in this pest [49]. Thus, our findings, in combination with this previous work, suggest that *S. marcescens* may be an important insect symbiont with the capacity to modulate host fitness across insect diversity. Our analysis of field strains of *N. lugens* also shows that this bacterium can be found across populations of a single strain, suggesting it can provide strong fitness benefits in pest insects that are exposed to intensive insecticide selection.

Multiple microorganisms from the soil have been identified that can degrade buprofezin, and putative biodegradation pathways have been described for some of these microbes [17,43]. However, only in *R. qingshengii* YL-1 has a gene cluster, *BfzBA3A4A1A2C*, been identified that encodes key proteins responsible for the upstream catabolic pathway of buprofezin [20] Specifically, the *BfzA3A4A1A2* cluster encodes a Rieske nonheme iron oxygenase (RHO) system that is responsible for the dihydroxylation of buprofezin [20]. In the *Bup_Serratia* genome, we identified three genes *NAE95_20165*, *NAE95_20175* and *NAE95_20145* with homology to *BfzA3A4A1A2*. *NAE95_20165* was predicted to encode an aromatic ring-hydroxylating dioxygenase subunit alpha, *NAE95_20175* was predicted to encode a [2Fe-2S] type ferredoxin and *NAE95_20145* was predicted to encode a GR-Type ferredoxin reductase (S9 and S10 Figs). According to a new classification scheme, NAE95_20165, NAE95_20175 and NAE95_20145 also constitute a D-IVα type RHOs [50]. RHOs are composed of one or two soluble electron transport proteins, viz. ferredoxins and reductases, and a terminal oxygenase [50]. The initial step in the biodegradation of aromatic compounds mediated by RHOs is the regio- and stereo-specific addition of both atoms of molecular oxygen into the aromatic nucleus of the substrate to form a cis-dihydrodiol [51]. This is consistent with the degradation of buprofezin in *R. qingshengii* YL-1 where *BfzA3A4A1A2* is responsible for the dihydroxylation of buprofezin at the benzene ring to produce buprofezindihydrodiol, which is the first step of buprofezin metabolism [20]. The α-subunit of RHOs contains two conserved domains, Rieske [2Fe-2S] center binding motif ($CXHX_{16-17}CX_2H$), which is known to function as an electron acceptor, and non-heme iron-binding motif ($E/DX_3DX_2HX_4H$), which is thought to be involved in oxygen binding [50]. These regions were conserved in NAE95_20165 (S11 Fig) suggesting it is a functional aromatic ring hydroxylating dioxygenase subunit. In contrast, the amino acid sequence of NAE95_20050, which also had homology to *BfzA1* of *R. qingshengii* and was annotated as a Rieske-type aromatic ring-hydroxylating oxygenase, does not have a fully conserved non-heme iron-binding motif (S11 Fig). This is relevant as Jiang et al. experimentally demonstrated that the activity of dioxygenases can be eliminated if conserved amino acids in such regions are changed [52]. Furthermore, phylogenetic analysis failed to group NAE95_20050 with other RHO system genes (Fig 4E). Together these findings suggest that NAE95_20050 may not be involved in the degradation pathway of buprofezin. Finally, our investigation of the expression of candidate RHO system genes in *N. lugens* in the presence of buprofezin, revealed that NAE95_20165, NAE95_20175 and NAE95_20145 might be responsible for the first step degradation of buprofezin in *Bup_Serratia*. Following buprofezin conversion into buprofezindihydrodiol by RHOs, the expected next step in its metabolism is a dehydrogenation reaction to generate dihydroxybuprofezin by a dehydrogenase. However, we only retrieved the NAE95_03695 annotated as aldo/keto reductase when we performed a blast search against the *Bup_Serratia* genome and proteome databases using BfzB dehydrogenase of *R. qingshengii* as the query. Although some dehydrogenases have an evolutionary convergence of function with aldo/keto reductase [53], no investigation has showed aldo/keto reductase have a dehydrogenation function. This is also consistent with our finding that *NAE95_03695* was not upregulated following exposure to buprofezin. In contrast, NAE95_20150, that has

sequence homogy with *R. qingshengii BfzC*, was upregulated (Fig 5). *BfzC* is predicted to encode a aromatic ring-cleaving dioxygenase that is responsible for the aromatic ring cleavage of dihydroxybuprofezin [20]. Therefore, it is possible that NAE95_20150 performs the same function in *Bup_Serratia* as with BfzC in *R. qingshengii*. These results, to some extent, also mirror their location in the *Bup_Serratia* genome with NAE95_20175, NAE95_20165, NAE95_20150 and NAE95_20145 found in close proximity in the genome (Fig 4D) suggesting they may be coregulated and share a generalized function. Based on our findings, we propose that the first step of buprofezin metabolism might be completed by the RHOs composed of NAE95_20165, NAE95_20175 and NAE95_20145 in *Bup_Serratia*, which is consistent with the situation in *R. qingshengii* YL-1. The aromatic ring-cleaved in the buprofezin degrading pathway is likely completed by the NAE95_20150. However, identification of the gene that is responsible for dehydrogenation before aromatic ring-cleavage requires further investigation.

The most commonly reported insecticide resistance mechanisms in insect pests involve changes in insecticide penetration through the cuticle, enhanced insecticide metabolism by insect detoxifying enzymes, and modification of the insecticide target site. However, these mechanisms are encoded by insect genomes, and often only evolve after many generations of insect exposure to insecticides [5]. Furthermore, these mechanisms have been frequently associated with fitness costs to the insect in the absence of insecticides [54–56]. These attributes provide opportunities to develop strategies that slow or manage the evolution of resistance. In contrast, symbiont-mediated insecticide resistance differs in a number of ways. Specifically: i) insects can acquire resistance to pesticides without any genomic changes and, potentially, associated fitness costs, (ii) the resistance trait can be acquired rapidly from symbionts in the environment, without the need for multiple generations of exposure and selection, and, (iii) symbiont-mediate resistance can be horizontally transmitted between individual insects and/ or strains, facilitating its rapid spread [5]. These properties mean that resistance evolution mediated by symbionts should be considered a serious threat to the sustainable control of damaging crop pests. Future research is thus urgently required to develop strategies to slow, prevent or overcome the development of this form of resistance. As demonstrated by our findings, such strategies should consider the use of antimicrobial treatments in combination with insecticides and/or exploiting the bacterial gene targets that encode the capacity to detoxify insecticides.

Finally, it would be interesting, in future, to examine the interaction between symbiont-mediate resistance mechanisms and those encoded by the insect host genome to understand if they combine additively or epistatically to yield the resistance phenotype. Investigation of potential mechanisms of resistance encoded in the genome of the NIB strain used in this study would be a first step towards this goal.

## Conclusion

Mutualism between microbes and insects plays a critical role in the growth and development of many insects, especially those feeding on nutrient-poor food sources such as plant sap [57]. Our study, in combination with a growing body of work, reveals that symbionts can also protect the insect host from pesticides [58]. As is the case for *S. marcescens*, these bacteria may be widely prevalent in the environment. Thus, exposure of bacteria to insecticides in agri-environments, and the rapid life cycle and large population sizes of these microbes, has the potential to rapidly select for bacterial strains that can more effectively metabolize insecticides leading to more potent resistance when acquired by pest insects. Our findings thus illustrate the importance of considering microbe—host insect—environment interactions in the development of integrated pest management strategies.

## Supporting information

**S1 Fig. Comparison of alpha diversity (sob chao1 ACE) of the microbiome of the NIS and NIB strains of *N. lugens*. $p > 0.05$, No significant difference (Kruskal-Wallis test).**
(TIF)

**S2 Fig. The sensitivity changes of *N. lugens* to buprofezin after *E. coli* was incubated to NIS strain. There is no significant difference. Difference comparisons were performed in GraphPad 7.0 with the log-rank test.**
(TIF)

**S3 Fig. Amino acid alignment of NAE95_20165 from different bacteria.** The red boxes circle different amino acides. A total of seven amino acides of NAE95_20165 from *Bup_Serratia* are inconsistent with other three *S. marcescens*.
(TIF)

**S4 Fig. Amino acid alignment of NAE95_20050 from different bacteria.** There are very low identities in NAE95_20050 among different *S. marcescens* strains, even NAE95_20050 is absent on genome of *S. marcescens* CM2012-028.
(TIF)

**S5 Fig. Amino acid alignment of NAE95_20175 from different bacteria.** It is completely conserved among different *S. marcescens* strains.
(TIF)

**S6 Fig. Amino acid alignment of NAE95_20145 from different bacteria.** The red boxes circle different amino acides. A total of twelve amino acides of NAE95_20145 from *Bup_Serratia* are inconsistent with other three *S. marcescens*.
(TIF)

**S7 Fig. Amino acid alignment of NAE95_20150 from different bacteria.** The red boxes circle different amino acides. A total of nine amino acides of NAE95_20150 from *Bup_Serratia* are inconsistent with other three *S. marcescens*.
(TIF)

**S8 Fig. Amino acid alignment of NAE95_03695 from different bacteria.** The red boxes circle different amino acides. A total of eighteen amino acides of NAE95_03695 from *Bup_Serratia* are inconsistent with other three *S. marcescens*.
(TIF)

**S9 Fig. Phylogenetic and conserved domains analysis of amino acides of ferredoxins from different bacteria, revealing NAE95_20175 belongs to [2Fe-2S] type ferredoxin.**
(TIF)

**S10 Fig. Phylogenetic of amino acides of ferredoxins reductase from different bacteria reveal NAE95_20150 belongs to GR-Type reductase.**
(TIF)

**S11 Fig. Amino acid alignment of oxygenase genes from different bacteria.** The two red boxes indicate the conserved Rieske [2Fe-2S] center and mononuclear iron-binding site respectively.
(TIF)

**S1 Table. Primers used in this study.**
(DOCX)

**S2 Table. Collecting information of *N. lugens* field populations.**
(DOCX)

**S3 Table. Summary of the sequencing data for the microbiota from the NIS and NIB strains of *N. lugens*.**
(DOCX)

**S4 Table. General features of the genome of *Bup_Serratia*.**
(DOCX)

**S5 Table. Gene information used for phylogenetic tree construction.**
(DOCX)

## Acknowledgments

We are grateful to Xin Yan (College of Life Science, Nanjing Agricultural University) for providing suggestions in experiment design and Guangzhou Genedenovo Biotechnology Co., Ltd for assisting in sequencing and bioinformatics analysis.

## Author Contributions

**Conceptualization:** Bin Zeng, Shun-Fan Wu, Chris Bass, Cong-Fen Gao.

**Data curation:** Bin Zeng, Ya-Ting Liu.

**Formal analysis:** Bin Zeng, Chris Bass.

**Funding acquisition:** Bin Zeng, Cong-Fen Gao.

**Investigation:** Bin Zeng, Fan Zhang, Ya-Ting Liu.

**Methodology:** Bin Zeng, Ya-Ting Liu, Shun-Fan Wu.

**Project administration:** Bin Zeng, Cong-Fen Gao.

**Software:** Bin Zeng.

**Supervision:** Chris Bass, Cong-Fen Gao.

**Validation:** Bin Zeng.

**Visualization:** Bin Zeng, Fan Zhang, Ya-Ting Liu.

**Writing – original draft:** Bin Zeng.

**Writing – review & editing:** Bin Zeng, Shun-Fan Wu, Chris Bass, Cong-Fen Gao.

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
