## [Decision Letter · Decision Letter 0]

5 Oct 2023

Dear Dr Gao,

Thank you very much for submitting your manuscript "Symbiotic bacteria confer insecticide resistance by metabolizing buprofezin in the brown planthopper, Nilaparvata lugens (Stål)" for consideration at PLOS Pathogens. As with all papers reviewed by the journal, your manuscript was reviewed by members of the editorial board and by several independent reviewers. In light of the reviews (below this email), we would like to invite the resubmission of a significantly-revised version that takes into account the reviewers' comments.

While most of the comments by the two reviewers relate to issues with clarity reviewer one has asked for an additional control. Please attempt to address all issues raised by the two reviewers.

We cannot make any decision about publication until we have seen the revised manuscript and your response to the reviewers' comments. Your revised manuscript is also likely to be sent to reviewers for further evaluation.

Sincerely,

Elizabeth A McGraw, PhD

Academic Editor

PLOS Pathogens

Ronald Swanstrom

Section Editor

PLOS Pathogens

Kasturi Haldar

Editor-in-Chief

PLOS Pathogens

orcid.org/0000-0001-5065-158X

Michael Malim

Editor-in-Chief

PLOS Pathogens

orcid.org/0000-0002-7699-2064

Dear Authors,

While most of the comments by the two reviewers relate to issues with clarity reviewer one has asked for an additional control. Please attempt to address all issues raised by the two reviewers.

Reviewer's Responses to Questions

**Part I - Summary**

Reviewer #1: Insecticide resistance in insect pest populations is one of major challenges in insect pest management. Understanding the mechanisms of insecticide resistance in insect pest populations is critically important for successful management of the pest populations. This study addresses an important question regarding the mechanism of buprofezin resistance in the brown planthopper, a major insect pest of the rice crop. The authors used many different approaches to address this research question. All the results are convincing and supportive to their conclusions. Although the contribution of symbiotic bacteria to insecticide resistance of the host insects has been reported by other researchers, this study definitely is unique in terms of this particular insect species potentially using buprofezin as a carbon source.

Reviewer #2: Zeng et al reported here a novel N. lugens resistance mechanism to buprofezin and emphasized the significance of factoring in the interactions among symbiotic bacteria, host insects and insecticides in the environment. They utilized a range of methods, including bioassays, amplicon sequencing, bacterial culture and inoculation, genomic analysis, and more, to provide compelling evidence that the symbiotic Serratia marcescens inside N. lugens confers buprofezin resistance through biodegradation. These findings hold substantial promise, both in the realm of scientific progress and in terms of pest management.

**Part II – Major Issues: Key Experiments Required for Acceptance**

Reviewer #1: I don't see any major issues regarding this study. However, I feel that there is room to improve the clarity of the manuscript as noted below:

1) The resistant strain originally collected from the field was 44.3-fold resistance to buprofezin as compared with the susceptible strain. After the laboratory selections with buprofezin for five generations, what was the level of buprofezin resistance of the insect when it was used in the study?

2) The authors mentioned "where buprofezin was used as the sole carbon source" Lines 200-201). This statement may be true when the authors only considered the chemical added to the LB medium; however, the statement seems to have ignored the carbon source from the LB medium, which the bacteria can use for them to grow. In comparison to the small amount of buprofezin in the medium, the major carbon source for the bacteria is the LB medium.

3) From the evolution point of view, I don't think that this particular symbiotic bacterial strain must rely on buprofezin as the sole carbon source for its growth as buprofezin as an insecticide is very new to the bacterium. If this is the case, the bacterium would not be able to survive on the LB medium without buprofezin.

4) It was a great idea to examine possible involvement of Bup_Serratia in degrading buprofezin by inoculating the buprofezin-susceptible planthoppers with Bup_Serratia followed by buprofezin bioassays. However, the results would be more convincing if the authors have included an additional control of the LB medium containing a S. marcescens strain incapable of degrading buprofezin. This additional control can eliminate possible absorbance of the insecticide by the bacterial cells or the cell debris, which reduce the contact of the insect to the insecticide. This is particular helpful as the difference in survival rate between the control (the LB medium only) and the treatment is only abut 20% at 120 hours (Fig. 3D).

5) Fig. 1B shows that the resistant strain became more susceptible (the survival rate decreased from about 80% to 40%) when the insect was treated with tetracycline. I was wondering if this relatively small change can completely explain over 40-fold of resistance. Are there any other resistance mechanisms involved in buprofezin resistance in this resistant strain, and/or is the growth inhibition of the bacteria after treated with the antibiotic under the experimental conditions complete. The authors might want to clarify or discuss this.

Reviewer #2: (No Response)

**Part III – Minor Issues: Editorial and Data Presentation Modifications**

Reviewer #1: 1) The manuscript needs to be further polished to correct some English issues. For example, on Line 24, a symbiotic "bacteria" should be changed to a symbiotic "bacterium". On Line 27, "sensitive" should be changed to "susceptible". On Line 59, the sentence "Among the most damaging of these pests and pathogens are insects" is grammatically incorrect.

2) The statement on Lines 41-43 is biased regarding different resistance mechanisms. Specifically, the alteration of the expression of detoxification enzymes is not caused by gene mutations. There are some examples that the gene mutations may result in more efficient enzymes for degrading insecticides.

3) On Line 51, I suggest to change "microbe x host insect x environment interactions" into "microbe-host insect-environment interactions".

4) On line 336, what do "Shannon and Simpson indices" exactly tell us? Please explain the differences.

5) Please describe the experiments of treating the NIS strain with the LB media containing Bup_Serratia. For example, how did the authors treat the insects and what was the time interval between the treatments of Bup_Serratia and buprofezin.

6) Fig3. D and E appear to be incorrectly labeled. Based on the data, E should be D while D should be E.

Reviewer #2: 1. The strains (NIS or NIB) used in each panel should be indicated in the Fig 1 for clarity.

2. I think the authors did not test the original G0 strains with antiboitics or 16S sequencing, so the buprofezin resistance may be associated with symbiont bacteria, but not "the increase of buprofezin resistance" . It will be also useful if the authors can give the rationale about why they chose this Yunan strain.

3, Fig 3D, what is SS? I prefer full name here. Besides, the figure legends for panel D and E should be exchanged.

4, From line 372 and in discussion, there is currently no biochemical or genetic confirmation regarding these candidate genes, and their amino acid sequence identity to known BfzA-C is not very high, thus the authors should exercise caution when drawing their conclusions.

5, The detection of Bup_Serratia from field planthopper populations is important to show the ecological relevance of this symbiont-mediated resistance. As a result, are these strains more resistant than NIS strains against buprofezin?

6, Line 81, the authors mentioned that there are other resistant mechanisms such as overexpression of P450 and target-site mutation, therefore, they should also check these genetic characteristics in both sensitive NIS strain and resistant NIB strain (because these two strains were collected from two distant locations). Such results will be helpful to show the precise contribution of symbionts to buprofezin resistance.

PLOS authors have the option to publish the peer review history of their article (what does this mean?). If published, this will include your full peer review and any attached files.

Reviewer #1: No

Reviewer #2: No
---

## [Editor Report · Decision Letter 1]

15 Nov 2023

Dear Dr Gao,

We are pleased to inform you that your manuscript 'Symbiotic bacteria confer insecticide resistance by metabolizing buprofezin in the brown planthopper, Nilaparvata lugens (Stål)' has been provisionally accepted for publication in PLOS Pathogens.

Best regards,

Elizabeth A McGraw, PhD

Academic Editor

PLOS Pathogens

Ronald Swanstrom

Section Editor

PLOS Pathogens

Kasturi Haldar

Editor-in-Chief

PLOS Pathogens

orcid.org/0000-0001-5065-158X

Michael Malim

Editor-in-Chief

PLOS Pathogens

orcid.org/0000-0002-7699-2064
---

## [Editor Report · Acceptance letter]

24 Nov 2023

Dear Dr Gao,

We are delighted to inform you that your manuscript, "Symbiotic bacteria confer insecticide resistance by metabolizing buprofezin in the brown planthopper, Nilaparvata lugens (Stål)," has been formally accepted for publication in PLOS Pathogens.

Best regards,

Kasturi Haldar

Editor-in-Chief

PLOS Pathogens

orcid.org/0000-0001-5065-158X

Michael Malim

Editor-in-Chief

PLOS Pathogens

orcid.org/0000-0002-7699-2064